# Assessment of Clinical Profile and Treatment Outcome in Vaccinated and Unvaccinated SARS-CoV-2 Infected Patients

**DOI:** 10.3390/vaccines10071125

**Published:** 2022-07-15

**Authors:** Ganesh Korishettar, Prashanth Chikkahonnaiah, SubbaRao V. Tulimilli, Siva Dallavalasa, Shashidhar H. Byrappa, SubbaRao V. Madhunapantula, Ravindra P. Veeranna

**Affiliations:** 1Department of Pulmonary Medicine, Mysore Medical College and Research Institute (MMC&RI), Mysuru 570001, Karnataka, India; ganeshsk47@gmail.com; 2Center of Excellence in Molecular Biology and Regenerative Medicine (CEMR) Laboratory (DST-FIST Supported Center), Department of Biochemistry (DST-FIST Supported Department), JSS Medical College, JSS Academy of Higher Education and Research (JSS AHER), Mysuru 570004, Karnataka, India; tulimillivenkatasubbarao@jssuni.edu.in (S.V.T.); sivadallavalasa@jssuni.edu.in (S.D.); mvsstsubbarao@jssuni.edu.in (S.V.M.); 3Department of Biochemistry, Council of Scientific and Industrial Research (CSIR)—Central Food Technological Research Institute (CFTRI), Mysuru 570020, Karnataka, India; shbmysore@gmail.com; 4Leader, Special Interest Group in Cancer Biology and Cancer Stem Cells (SIG-CBCSC), JSS Medical College, JSS Academy of Higher Education and Research (JSS AHER), Mysuru 570004, Karnataka, India; 5Department of Pathology, Mysore Medical College and Research Institute (MMC&RI), Mysuru 570001, Karnataka, India

**Keywords:** SARS-CoV-2, COVID-19, vaccinated, nonvaccinated, cross-sectional prospective cohort study

## Abstract

Vaccines against severe acute respiratory syndrome-corona virus-2 (SARS-CoV-2) infection, which causes coronavirus disease–19 (COVID-19) in humans, have been developed and are being tested for safety and efficacy. We conducted the cross-sectional prospective cohort study on 820 patients who were positive for SARS-CoV-2 and were admitted to Princess Krishnajammanni trauma care centre (PKTCC), Mysore, which was converted to a designated COVID hospital between April 2021 to July 2021. After obtaining the informed consent, RT-PCR report, vaccination certificate and patient history, patients were classified according to their vaccination status. Results from the study showed decreases in serum ferritin levels, clinical symptoms, improvement in oxygen saturation, early recovery in patients having diabetes and hypertension, and a substantial reduction in the overall duration of hospital stay in vaccinated patients compared to unvaccinated patients. Further, fully vaccinated patients showed better outcomes compared to single dose vaccinated and nonvaccinated patients. Taken together, our findings reaffirm the vaccine’s effectiveness in reducing case fatality and promoting faster recovery compared to nonvaccinated patients. Efforts to increase the number of immunized subjects in the community help to achieve herd immunity and offer protection against the severity of COVID-19 and associated complications while minimizing the public health and economic burden.

## 1. Introduction

The new public health crisis, known as the 2019 novel coronavirus (2019-nCoV) or the severe acute respiratory syndrome coronavirus 2 (SARS-CoV-2), continues to threaten the entire world with its rapidity in spreading and severity in causing mortality. SARS-CoV-2 virus, discovered in bats, is reported to be transmitted to humans via unidentified intermediary species in the city of Wuhan, Hubei Province, China [1]. Infection with SARS-CoV-2 causes severe respiratory problems, collectively referred to as coronavirus disease-19 (COVID-19). The severity of the COVID-19 pandemic is a major concern as it infected 550 million people and caused 6.32 million deaths across the globe [2]. The top five countries with the highest numbers of reported cases are the United States of America (85673420), India (43362294), Brazil (31890733), France (29416740), and Germany (27681775). The top five countries with the highest number of reported deaths are the United States of America (1005025), Brazil (669530), India (524954), the Russian Federation (380776), and Mexico (325487) [2]. Epidemiological studies revealed that elderly patients are more vulnerable to developing COVID19 compared to children and teenagers [3,4].

The SARS-CoV-2 is an enveloped virus possessing a positive sense single stranded RNA of approximately 30 Kb in length [5,6]. The virus is made up of several proteins involved in mediating the virus infectivity and spread [7,8]. The spike (S), envelope (E), membrane (M), and nucleocapsid (N) proteins constitute the key proteins in the virus [7,8]. The virus enters the cell through a receptor-mediated endocytosis process releasing the virus genome into the cytosol [9,10,11,12]. Once in the cytosol, the SARS-CoV-2 begins to express various viral proteins by rapid replication and translation of its genomic RNA. The newly produced RNA and proteins assemble to produce new viral particles [13]. Patients infected with SARS-CoV-2 experience mild to moderate flu-like symptoms such as fever, cough, diarrhoea, muscle ache, and breathing difficulties [14,15]. In the case of severe disease, the patients suffer from acute systemic inflammation described as the “cytokine storm”, causing respiratory and multiorgan failure [14,15]. In addition to symptomatic patients, asymptomatic patients can also shed the virus, contributing to the rapid spread of the virus across the globe [14,15].

SARS-CoV-2 infection is detected by a reverse transcription polymerase chain reaction (RT-PCR) test, which detects the viral RNA in the patient sample [16]. In addition, many other antibody tests, such as the rapid-antigen test, have also been developed and are available on the market for detecting the viral infection [17]. Among these various COVID-19 diagnosis tests, RT-PCR is considered a gold-standard test for detecting the SARS-CoV-2 infection [18]. In some studies, the results of RT-PCR have also been used to determine the severity of infection as the authors have observed a correlation between cycle threshold (Ct) value and viral load [19,20,21]. However, due to several concerns and queries about the use of the Ct value for determining the disease severity, the Ct value was subsequently considered only for measuring the viral load [22].

Several academic institutions and vaccine production industries-initiated research on the development of vaccines against SARS-CoV-2 infection as soon as the virus genome was published in early January 2020 [23]. According to WHO, 354 COVID-19 vaccine candidates have been developed and are currently in various stages of development [24]. Recent statistics have revealed that among these vaccine candidates, 156 are in the clinical stage of development, whereas the remaining 198 are in preclinical stages [24]. As of 13 May 2022, thirty-eight vaccines of different categories have been authorized in at least one country for combating the severity of COVID-19 [25]. Two vaccines were approved initially for use in India. The COVAXIN, which was developed by the Indian Council of Medical Research (ICMR) and Bharat Biotech, Hyderabad [26,27] and the COVISHIELD, which was developed by AstraZeneca/Oxford University and manufactured by the Serum Institute of India (SII) [28,29]. Between two vaccines, COVISHIELD was widely available during our study period (April 2021 to July 2021). COVISHIELD is a recombinant adenovirus vector vaccine encoding Spike (S) glycoprotein [30]. Numerous trials have been conducted to evaluate the safety and efficacy of COVISHIELD, which found that the vaccine is safe, immunogenic, and effective in preventing symptomatic and severe diseases [31,32]. The majority of COVID-19 symptoms were much less common in vaccinated patients compared to unvaccinated patients [33]. In addition, most patients in the vaccinated groups (single dose or double dose) are completely asymptomatic and experience infection-associated severity very minimally [34].

Mortality due to COVID-19 has been reported to be relatively high in aged (60 years and above) patients [35], and in particular, those individuals with comorbidities such as diabetes, hypertension, cardiovascular disease, cancer, etc. [36,37,38]. Therefore, such individuals were categorized as high-risk groups and were administered vaccines on priority. This group was followed by individuals between 45 years to 59 years of age and subsequently by the younger group. Despite the high level of vaccine efficacy, a small percentage of fully vaccinated patients were infected with SARS-CoV-2 and exhibited symptomatic COVID-19 features, and in very few cases that are associated with comorbidities succumbed to death [39]. Hence, the present study was conducted to determine the clinical features and outcome of COVID-19 among vaccinated and nonvaccinated patients admitted to a designated COVID hospital in Mysore in southern India.

## 2. Material and Methods

**Study design, ethical consideration, data collection, and case enrolment:** A cross-sectional cohort study was conducted among patients who were diagnosed with COVID-19 either by Rapid Antigen Test or by RT-PCR (VIRALDTECT–II Multiplex Real Time PCR Kit for COVID-19, Genes2Me, Gurugram, Haryana, India), which was approved by Indian Council of Medical Research (ICMR), New Delhi, Government of India. These patients were admitted to a designated COVID hospital in Mysuru, Karnataka, India. After taking prior approval from the institutional ethics committee of Mysore Medical College and Research Institute (MMC&RI) and Associated Hospitals, Mysore (EC REG ECR/134/INST/KA/2013/RR-19), written consent, RT-PCR test report, vaccination certificate issued by the Govt. of India, the data related to the study, which included oxygen saturation at the time of admission, clinical features, and comorbidities were collected. Since COVISHIELD was the widely available vaccine during our study period (April 2021 to July 2021), all of the patients who received the vaccine were administered with COVISHILD. Based on the vaccination status, patients were classified into unvaccinated and vaccinated, including single dose vaccinated or fully vaccinated group. The patients were followed until discharge, and outcomes such as oxygen saturation, serum ferritin, clinical conditions, and duration of stay in hospital were assessed.

**Study sites and sample size:** The study was carried out by the Department of Respiratory Medicine, MMC&RI, at Princess Krishnajammanni Trauma Care Centre (PKTCC), which was converted into a designated COVID hospital. The study subjects included both male and female patients admitted to PKTCC during the study period. 

**Exclusion criteria:** Patients who were admitted with COVID-like syndrome with symptoms such as headache, muscle pain, fever, cough, etc., and who tested negative for COVID-19 by RT-PCR, and patients who were referred to other hospitals were excluded from the study.

**Estimation of serum ferritin:** Serum ferritin level was estimated by chemiluminescence method in a COBAS 6000 from Roche Diagnostics, Indianapolis, IN, USA. The method works on a sandwich principle wherein a ferritin-specific antibody and a labeled ferritin-specific antibody form a sandwich complex in the first step. In the 2nd step, the microparticles were added to enable sandwich binding to the solid phase. Subsequently, the reaction mixture was aspirated into the measuring cell, where microparticles were captured onto the electrode. Following the removal of unbound microparticles, a voltage was applied, resulting in induction chemiluminescence. The emitted chemiluminescence was measured by a photomultiplier, and the obtained readings were used to quantify serum ferritin using a calibration curve.

**Statistical Analysis:** The findings in this study are descriptive and inferential statistical in nature. The results were analyzed using SPSS version 20.0 (IBM Corporation, SPSS Inc., Chicago, IL, USA). Results on continuous measurements were presented on Mean ± SD (Standard Deviation), and results on categorical measurements were presented in frequency (percentage). A *p*-value of <0.05 and 0.0001 was considered statistically significant.

## 3. Results

Between April to July 2021, the study was conducted among 820 patients who were admitted to PKTCC. They were segregated as unvaccinated, single dose vaccinated, and as fully vaccinated (two doses), and the outcomes were compiled according to their clinical features, saturation at the time of admission, laboratory parameters, vaccination status, and duration of stay in hospital. The vaccination status was ascertained based on the vaccination certificate produced by each patient at the time of admission. Further, the certificate’s authenticity was also verified based on the QR code printed on the certificate. During the study period, we observed that all those patients in the vaccinated group were administered with COVISHIELD, perhaps due to the wide availability and acceptability of the COVISHIELD. Among 820 patients, 626 were unvaccinated, 147 were single dose vaccinated, and only 47 were fully vaccinated. A flow chart showing the overview of the study and classification of the patients is shown in Figure 1.

### 3.1. Gender and Age Distribution

A total of 520 male patients (63.41%) were in the study. The male-to-female sex ratio was 1.73:1. The mean age of the patients was 50.55 ± 5.35 years. The majority of the patients (65%) belonged to the 31–60 years age group, with more than 20% of patients aged above 60 years, while only <15% of patients were in the 18–30 years age group. (Figure 2).

### 3.2. Oxygen Saturation

In this study, the mean oxygen saturation in patients at the time of admission was 89%, 92%, and 95% in unvaccinated, single dose vaccinated, and fully vaccinated patients, respectively. Further, the mean oxygen saturation in fully vaccinated and single dose vaccinated patients was statistically significant compared to unvaccinated patients, with *p*-value < 0.05 and <0.0001, respectively (Figure 3).

### 3.3. Serum Ferritin Levels

In this study, we have quantified the serum ferritin level among vaccinated and unvaccinated patients. Analysis of the data showed that the mean serum ferritin level was significantly higher (*p* < 0.0001) in unvaccinated (665.85 ± 557.70 ng/mL) patients by more than two-fold compared to vaccinated patients. Patients who had received their single dose of the vaccine had decreased mean serum ferritin levels (412.23 ± 421.32 ng/mL), while patients who had received full vaccination had ferritin levels (282 ± 349.88 ng/mL), which was very close to normal healthy volunteers (Figure 4).

### 3.4. Clinical Features

Results obtained from this study showed that the most common symptom in COVID-19 patients is cough. About 78.4% of patients in the unvaccinated group had severe cough, compared to approx. 77.7% and approx. 74.5% in single and fully vaccinated patients, respectively. Breathlessness was reported in about 54.0% of unvaccinated patients. However, about 50.5% and 40.4% of single and fully vaccinated patients reported breathlessness. Fever was reported by about 72.7% of patients in the unvaccinated group, which was slightly higher compared to patients who had received a single dose (68.9%) or were fully vaccinated (70.2%) patients. COVID-19 patients reported loose stools in about 2.4% of the unvaccinated, and 1.9% to none in single and fully vaccinated patients, respectively. A total of 0.7% of unvaccinated patients reported haemoptysis compared to none in both single dose and fully vaccinated patients. Vomiting was reported in about 1.3% of patients who had not received any vaccines; however, 3.9% and 4.3% of single or fully vaccinated patients reported vomiting (Figure 5). The reason for this discrepancy was not known.

### 3.5. Average Hospital Stay

Next, we evaluated the average hospital stay among the vaccinated and unvaccinated patients. As shown in Figure 6, the average duration of hospital stay was significantly less (5.65 ± 1.01 days) in fully vaccinated patients. Patients who had received a single dose of vaccine had an average hospital stay of 7.65 ± 1.38 days, while the hospital stay was much high for patients who were not vaccinated (9.43 ± 2.54 days).

### 3.6. Comorbidities

Similarly, in our study, diabetes and hypertension were the most prevalent comorbidities, with 28.41% and 22.92% of patients reporting diabetes and hypertension, respectively. Around 2% reported cardiac comorbidities in this study (Figure 7). The recovery rate was substantially higher in fully vaccinated (89.89%) followed by single dose vaccinated (79.3%) and unvaccinated (74.9%) patients. On the other hand, the death rate was higher in unvaccinated (12.8%) compared to single dose vaccinated (8.3%) and fully vaccinated (8.7%) patients (Figure 7). 

## 4. Discussion

COVID-19 is one of the major pandemics ever seen by humans [40]. The disease-causing virus SARS-CoV-2 spread across various parts of the world and killed millions of patients, particularly those over 60 years of age [41]. Due to its ability to bind to respiratory tissues, cause cytokine storms, and spread all over the body, the virus has caused not only severe respiratory illness but also damaged several other organs in infected patients [42]. As a result, the infected patients suffered from various complications other than respiratory problems, even after recovery from infection [43]. Therefore, preventive measures, including social distancing, masking, and vaccination, have helped to slow down the virus transmissibility. [44]. Although vaccination has been found to protect patients, several recent studies have reported the occurrence of infection even in fully vaccinated patients, raising concerns about the efficacy and duration of protection provided by the vaccines [45,46]. Therefore, it was important to test whether vaccination (single dose or double dose) has beneficial health effects compared to patients who have not received the COVID-19 vaccine. Hence, in this study, we have measured various clinical outcomes in COVID-19 patients who had received a single dose or two doses of vaccine and compared the effects with patients who had not received any vaccine shots. This is one of the largest studies done in the region, comprising 820 COVID-19 patients admitted to the hospital.

We found that vaccination helped in the faster recovery of patients who had received at least one dose of the vaccine compared to unvaccinated patients. The case fatality rate was substantially less in patients with diabetes and hypertension compared to single dose or unvaccinated patients, indicating a faster recovery. Among various comorbid conditions in vaccinated patients, patients suffering from hypertension recovered better. The overall outcome of this study with a single dose and double dose of administration of the COVID-19 vaccine was 79.3% and 89.89%, which is comparable with the data reported by Bernal (2021). About 36% and 30% of the overall outcome was reported by Bemal 2021 with a single dose of the BNT162b2 vaccine and ChAdOx1 nCoV-19 vaccine, respectively. The overall outcome was increased to 88% and 67% with two doses of the BNT162b2 vaccine or two doses of the ChAdOx1 nCoV-19 vaccine, respectively [47]. In summary, vaccination protects patients from SARS-CoV-2 infection and reduces the case fatality rate. Even though a fraction of vaccinated patients succumbed to death, the primary reason might not be the SAR-CoV-2 infection but could be due to underlying comorbidities or old age. Supporting this observation, many recent studies have published similar findings and reported that vaccination might not offer complete protection in patients with comorbidities or old age, as the host immune response to vaccination may not yield protective antibody titres [48]. However, several studies show that vaccination overall protects the patients, promotes recovery, and reduces the case fatality rate and COVID-19-associated complications [49].

Normal blood oxygen saturation levels in humans are 95–100%. Oxygen saturation below 90% is considered low and is called hypoxemia [50]. The National Institutes of Health (NIH), USA, recommended a target oxygen saturation range of 92–96% for patients with COVID-19 [51]. Patients with COVID-19 infections usually experience a low level of oxygen in their blood [52]. Several factors, including local and systemic inflammation, affect blood oxygen saturation [53]. Thus, the measurement of blood oxygen saturation is a standard screening tool in clinical settings to assess the performance of the respiratory system [54,55]. Low oxygen saturation and elevated respiratory rate have been consistently associated with disease severity in patients hospitalized with COVID-19 [56,57,58,59]. Measurement of oxygen saturation is one of the critical indications for monitoring patients with COVID-19. Since a drastic decrease in oxygen saturation, i.e., hypoxic condition, is fatal, especially in those with respiratory symptoms, it is important to monitor the oxygen saturation level regularly [60]. In COVID-19, many patients have reported low oxygen levels compared to uninfected patients [61]. Vaccination of patients helped in reducing the severity of the hypoxic condition. A retrospective analysis of 6180 patients infected with COVID-19 in the USA revealed significantly higher chances of death in patients with lower oxygen saturation and higher respiratory rate on admission [62]. In our study, mean oxygen saturation was 95% and 92% in fully vaccinated and single dose vaccinated patients, respectively. However, in unvaccinated patients, mean oxygen saturation was 89%, significantly lower (*p* < 0.05 and *p* < 0.0001) than in single and fully vaccinated patients. This indicates that vaccination protects the patients against the disease’s severity and helps maintain oxygen saturation. This indicates that vaccination protects the patients against the disease’s severity and helps maintain oxygen saturation. Lower oxygen saturation was one of the main reasons for admission of unvaccinated patients in hospitals compared to single dose or fully vaccinated patients. We observed a slight drop in oxygen saturation between single vs. fully vaccinated patients, though one of the very recent studies reported no significant differences in the blood oxygen saturation among vaccinated patients [63]. In another study, Wilder-Smith et al., 2022, reported that vaccination could reduce the risk of even delta variant infection and promote viral clearance [64].

The SARS-CoV-2 infection causes systemic inflammation in patients, especially with severe diseases. Elevated release of pro-inflammatory cytokines, e.g., IL-6, TNF-alpha, etc., caused cellular damage, metabolic acidosis, ROS generation, and secondary tissue damage [65,66,67,68]. Recent studies have reported elevated serum ferritin levels in critically ill COVID-19 patients [69]. Ferritin is an iron storage protein that regulates immune function and inflammation [70]. Ferritin is a key mediator of immune dysregulation, hence, contributing to the cytokine storm [71]. In addition, individuals with COVID-19 reported elevated serum ferritin levels [68]. A strong correlation between COVID-19 severity and elevated ferritin levels was reported in clinical settings. Ferritin levels were found significantly higher in patients with severe COVID-19 (2800 ng/mL) than in non-severe COVID-19 (708 ng/mL). Similarly, ferritin levels were directly associated with an increased risk of ARDS in patients with COVID-19 in an Italian study [72]. Furthermore, a separate study reported that COVID-19 patients with increased ferritin levels had a longer period for viral clearance, hence, longer hospital stay [73]. Similarly, another study reported a strong correlation between elevated ferritin levels and in-hospital mortality and invasive ventilator dependence [74]. A systematic review and meta-analysis highlighted serum ferritin as an important biomarker in the management of COVID-19. They concluded that high serum ferritin levels were found to be associated with more severe disease and negative/poor outcomes in COVID-19 [75]. In the present study, the serum ferritin levels were significantly higher in unvaccinated patients (665. 85 ng/mL), whereas the level decreased marginally in single dose vaccinated patients (412.23 ng/mL) and reached a normal status in fully vaccinated patients (282 ng/mL). A study reported that the fatal outcomes of COVID-19 are partly due to unusually high cytokines, collectively known as a “cytokine storm”. The severity of COVID-19 not only depends on the cytokine storm but also on the ferritin levels [76]. One of the studies reported that patients with very severe COVID-19 exhibited higher serum ferritin levels than severe COVID-19 individuals [71]. Gomez-Pastora et al. (2020) reported that ferritin concentration above approx. 400 ng/mL is a risk factor for progression to severe disease [77].

Major symptoms of COVID-19 are cough, breathlessness, fever, loose stool, and haemoptysis; which were reduced in the patients vaccinated with a single dose and fully vaccinated as compared to the unvaccinated patients. A multi-centric study with a sample size of 1446 patients conducted in four tertiary care hospitals in Kerala, South India, showed that severity of infection, duration of hospital stays, need for ventilation, and death were significantly less among vaccinated patients vs. unvaccinated patients [78]. Similarly, another retrospective cohort study conducted in Brazil reported the reduction in hospital stays and the need for mechanical ventilation in infected health care workers vaccinated with Oxford-AstraZeneca (ChAdOx1) or Corona vac [79]. Similar studies conducted elsewhere have shown the efficacy of vaccines in reducing the severity of COVID-19 and the length of hospital stay [80,81,82,83,84]. Similar to other studies, our study also shows a reduction in hospital stay by four days in fully vaccinated patients compared to unvaccinated patients indicating the efficacy of the COVISHIELD vaccine. 

Diabetes mellitus, hypertension, ischemic heart diseases, and chronic kidney diseases were the most common comorbidities reported by COVID-19 patients [85,86]. A recent study evaluating the possible reasons for the mortality following COVID-19 vaccination found that the deaths are due to pre-existing comorbidities and not due to the vaccination or the failure of vaccines to extend the protection [87]. A retrospectively analysed report showed that among patients hospitalised with a SARS-CoV-2 delta variant (B.1.617.2) infection, vaccination was associated with less severity, even in the presence of comorbidities [88]. In the present study, we observed that the recovery and death rate in vaccinated patients with comorbidities was better than the unvaccinated patients with comorbidities. The overall recovery in unvaccinated was 74.9%, whereas in single dose vaccinated patients, it was 79.3% and 89.89% in fully vaccinated patients. The death rate was 12.8% in unvaccinated patients, whereas it was 8.7% and 8.3% in single dose and fully vaccinated patients. In summary, vaccination helped protect the patients from acquiring reinfection with SARS-CoV-2 and also assisted in reducing the complications associated with COVID-19.

## 5. Conclusions

This cross-sectional cohort study which was conducted among 820 patients who were diagnosed positive for COVID-19 and admitted to the Princess Krishnajammanni Trauma Care Center in Karnataka state, India, suggests decreases in serum ferritin levels, improvement in oxygen saturation, decreases in clinical features (cough, breathlessness, fever, and loose stools), and average duration of hospital stay in vaccinated patients versus unvaccinated patients. Some deaths reported in vaccinated patients were primarily due to their underlying comorbidities or old age. This is one of the largest studies done in this region of the Karnataka State on COVID-19-positive patients and assumes significance for the fact that it provides the effectiveness of the COVISHIELD vaccine against SARS-CoV-2 infection and associated complications. Even though we report better outcomes and less severity in the vaccinated patients, the results of our study should be treated cautiously owing to the presence of various confounding factors at the individual level, including gender, age, presence or absence of comorbidities, diseases of the immune system, and the nature of the vaccine itself. Therefore, the current study requires that outcomes among unvaccinated, single dose, and fully vaccinated patients be adjusted for the above potential confounders. Our study reaffirms continued mass vaccination against COVID-19 to achieve overall herd immunity and protection against COVID-19.

## Figures and Tables

**Figure 1 vaccines-10-01125-f001:**
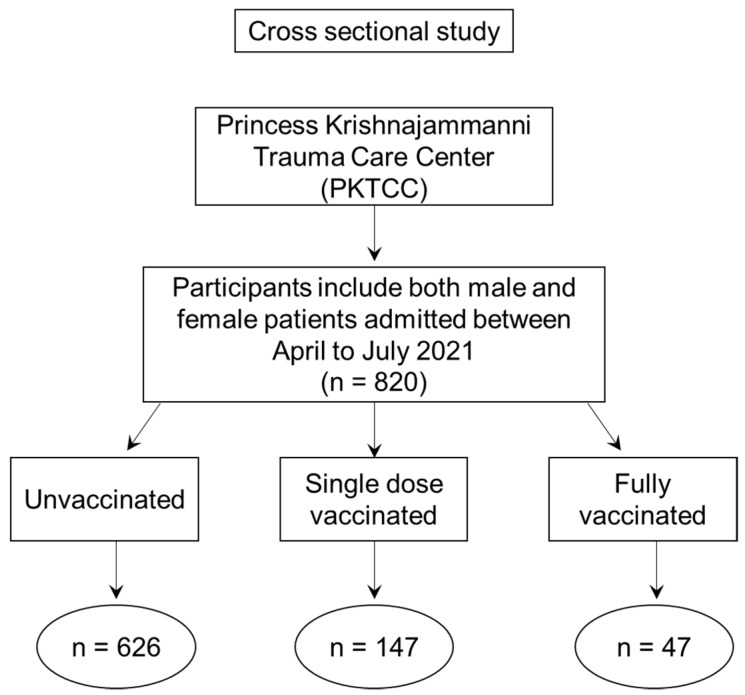
Flow chart showing the study overview and classification of the total number of patients.

**Figure 2 vaccines-10-01125-f002:**
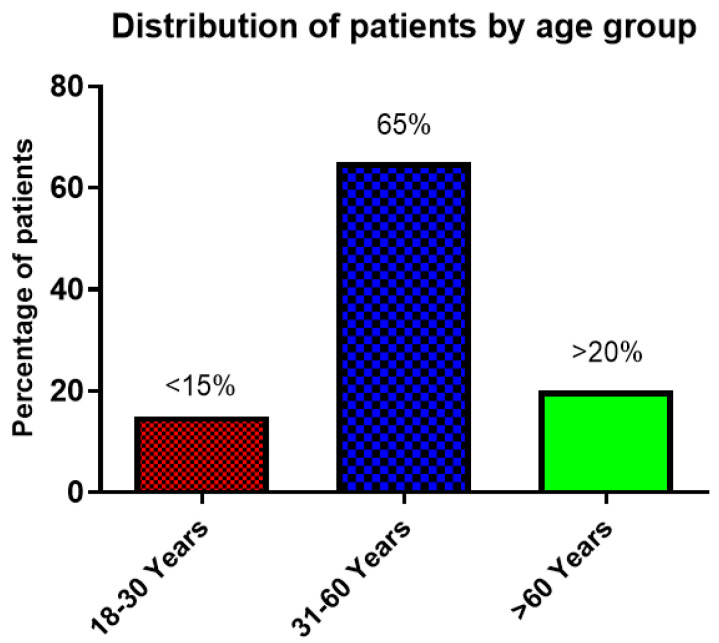
The graph shows the distribution of patients by age group.

**Figure 3 vaccines-10-01125-f003:**
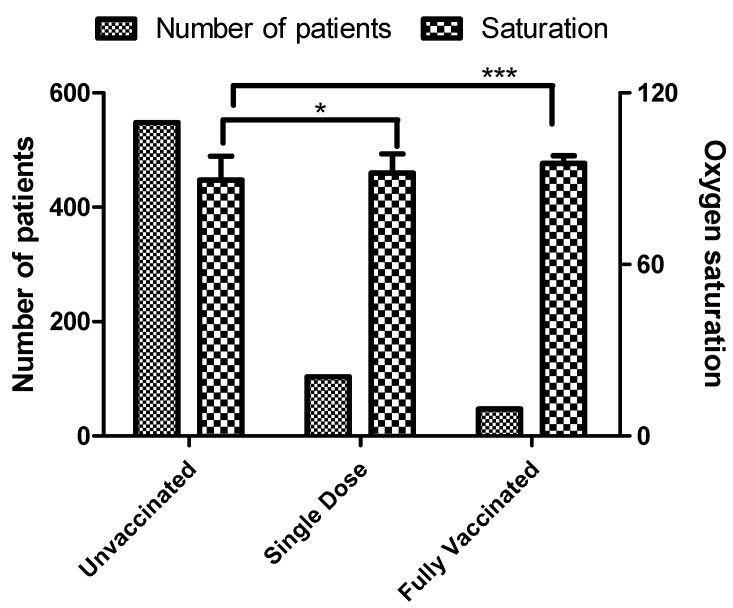
Oxygen saturation in patients at the time of admission. The bar diagram represents the mean oxygen saturation in patients. The data is represented in Mean ± SD (Standard Deviation). Multiple comparisons were performed using one-way ANOVA with Tukey’s Post Hoc Test. * and *** represent significance (*p*-value < 0.05 and <0.0001) in single dose and fully vaccinated patients compared with unvaccinated patients.

**Figure 4 vaccines-10-01125-f004:**
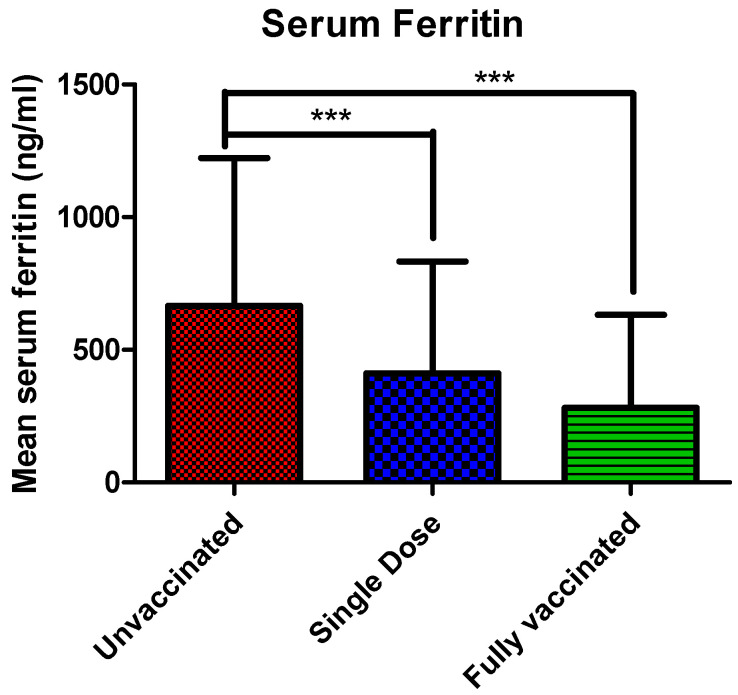
The serum Ferritin levels in unvaccinated, single dose, and fully vaccinated patients. The bar diagram represents mean serum ferritin levels (Mean ± SD) in admitted patients. Multiple comparisons were performed using one-way ANOVA with Tukey’s Post Hoc Test. *** represents a significant increase (*p* < 0.0001) in unvaccinated patients compared to vaccinated patients.

**Figure 5 vaccines-10-01125-f005:**
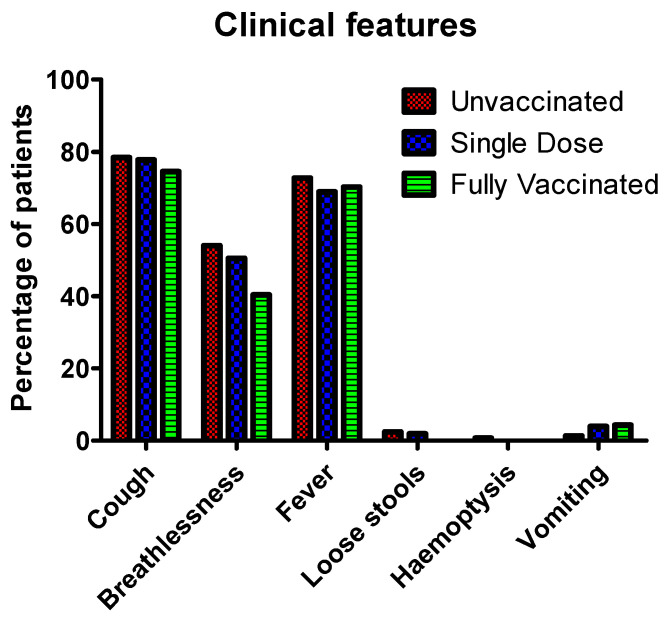
Distribution of clinical features among unvaccinated, single dose, and fully vaccinated patients.

**Figure 6 vaccines-10-01125-f006:**
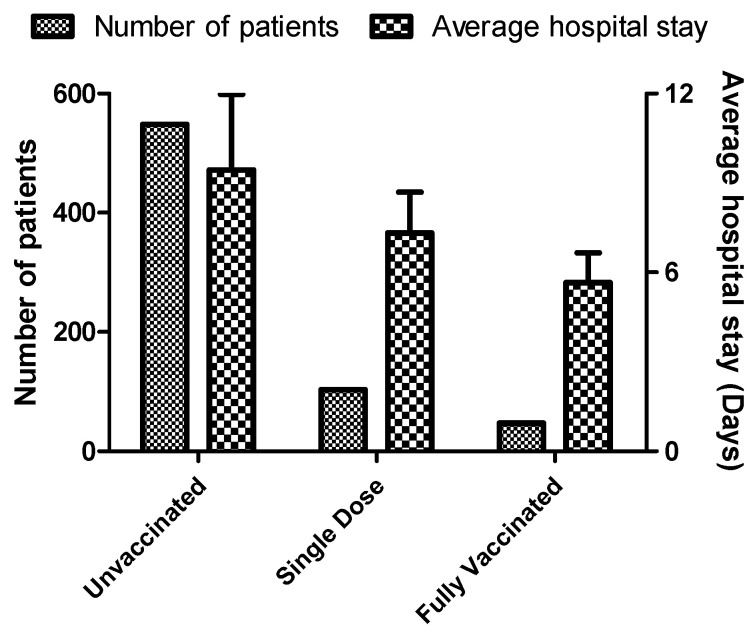
The average duration of hospital stays among vaccinated and unvaccinated patients. Data is represented in Mean ± SD.

**Figure 7 vaccines-10-01125-f007:**
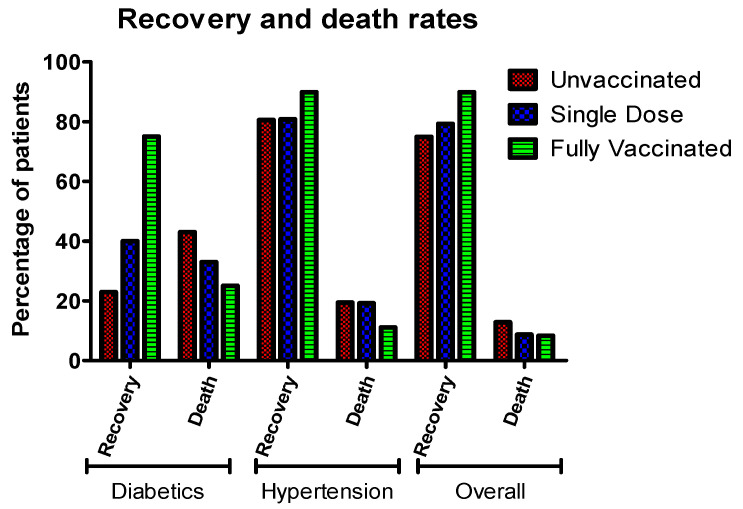
Overall outcome in patients with diabetes and hypertension.

## Data Availability

Data are available upon reasonable request. The dataset used to conduct the analyses is available from the corresponding author upon reasonable request.

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
