# Peer review of "Assessment of Clinical Profile and Treatment Outcome in Vaccinated and Unvaccinated SARS-CoV-2 Infected Patients"

_vaccines, 2022, doi:10.3390/vaccines10071125_

Round 1

Reviewer 1 Report

These data make a significant contribution to public health, with a large study addressing the important topic of the impact of vaccination at one or doses on benefits to hospitalized COVID-19 patients.  There are some statistics issues that need more work to make the results more rigorous, credible, and interpretable.

Statistical comments

1. Can the article elaborate how vaccination status was ascertained, and how we can be confident in the ascertainment?  Could there be misclassification?

2. Because this is a non-randomized/observational study, the results are susceptible to potential confounding, as noted in the Discussion.  However, the statistical analyses comparing outcomes among the unvaccinated, single dose, and double dose groups, did not adjust for potential confounders.  The article would be strengthened by including analyses that adjust for potential confounders (e.g., with linear regression).

4. lines 193-196: What are the +/- numbers?  They are not self-evidence, so what they are needs to be stated.  

5. Fig 3, Fig 4, Fig 6, there need to be 'uncertainty bars' that go both down and up, and the figure captions need to define the bars.  Are they 95% confidence intervals about means?  Or standard errors from the means?

6. lines 327-330: +-xx% meaning is not given/not self-evidence, please provide the detail about what is being shown.

Other comments

1. Line 53: '530 million infected', it must be many more than this, especially given estimates that 80% of infections are asymptomatic.

2. Lines 55-57: Provide a date for which the website was accessed for citing the counts.

3. Lines 83-84: Ct value and viral load copies/ml are essentially deterministically linked through a conversion formula. I would think the logical point to make here would be citing evidence that Ct value is associated with severity of disease.

4. Line 88: This sentence suggests vaccine efficacy against INFECTION, but really theses vaccines are not that efficacious against infection, they are efficacious against symptomatic disease.  That is true for essentially every good vaccine.  Would recommend, when referring to vaccine efficacy, always restricting to prevention of symptomatic disease; this was always the primary endpoint of the phase 3 trials.  Mary Lou Clements-Mann in a 1998 AIDS Res Hum Retroviruses paper is a classic commentary on how there is often confusion in that people talk about vaccine efficacy against infection, but really almost all vaccine efficacy trials study symptomatic disease endpoints, not infection endpoints.

5. Lines 104-105: rephrase to note disease reduction is the benefit, not infection reduction.  This issue comes up again in lines 202-203.

6. Line 155: "Patients who were admitted with COVID like syndrome"  Why were they excluded?  On the surface this sounds like excluding COVID-19 patients.

7. Lines 234-236: a typo in stating that 72.7% is 'much higher' than 68.9% and 70.2%

8. Lines 264-266: 'sequentially higher' is not accurate

Author Response

Dear Dr. Federico,

We sincerely thank the reviewer for their time and for their comments to improve the quality of our article. Please find the point-by-point response to each of the reviewer’s comments below. 

Reviewer-1

Statistical comments

Query #1: Can the article elaborate how vaccination status was ascertained, and how we can be confident in the ascertainment? Could there be misclassification?

Response: We thank the reviewer for this important comment. The vaccination status was ascertained based on the vaccination certificate produced by each patient at the time of admission. The vaccination certificate is issued to all individuals by the Govt. of India after the administration of each dose of vaccine. Further, the authenticity of the certificate was also verified based on the QR code printed on the certificate. We believe that there is no chance for misclassification. As per the suggestion of the reviewer, we have made changes to the manuscript. Lines: 168-173.

___________________________________________________________________________

Query #2: Because this is a non-randomised/observational study, the results are susceptible to potential confounding, as noted in the Discussion. However, the statistical analyses comparing outcomes among the unvaccinated, single-dose, and double-dose groups did not adjust for potential confounders. The article would be strengthened by including analyses that adjust for potential confounders (e.g., with linear regression).

Response: We thank the reviewer for the suggestion. We agree with the reviewer that results are susceptible to potential confounding factors, which are mainly age, gender, presence of comorbidities, asymptomatic, type of the vaccine, and dose-response in each patient etc. The linear regression analysis could help in normalising those variable factors. However, in our manuscript, we have analysed a total of 820 patients for some of those confounding factors individually. The analysis was further statistically analysed for the significance of their response to COVID-19 severity in unvaccinated, single-dose and double-dose vaccinated individuals.

___________________________________________________________________________

Query #3: lines 193-196: What are the +/- numbers? They are not self-evidence, so what they are needs to be stated.

Response: Authors would like to thank the reviewer for the comment. +/- indicates the Mean±SD (Std. deviation). This is used to designate the oxygen saturation of each patient at the time of admission to the hospital (Lines: 195-199).

___________________________________________________________________________

Query #4: Fig 3, Fig 4, Fig 6, there need to be 'uncertainty bars' that go both down and up, and the figure captions need to define the bars. Are they 95% confidence intervals about means? Or standard errors from the means?

Response: We would like to thank the reviewer for the comment to bring clarity to the readers. As per the suggestions, we have included the figure legends for all three figures [Lines 195-199-212 (Fig 3), 209-212 (Fig 4) and 238-239 (Fig 6)].

Query #5: lines 327-330: +-xx% meaning is not given/not self-evidence, please provide the detail about what is being shown.

Response: We thank the reviewer for the note. As per the suggestion, we have represented the mean oxygen saturation in absolute percentage, which is easy to follow by the readers. Accordingly, we have made changes to the manuscript. (Lines: 303-310)

“In our study, mean oxygen saturation was 95% and 92% in fully vaccinated, and single-dose vaccinated patients, respectively. However, in unvaccinated patients, mean oxygen saturation was 89%, significantly lower (p<0.05 and p<0.0001) than in single vaccinated and fully vaccinated patients. This indicates that vaccination protects the patients against the disease's severity and helps maintain oxygen saturation”.

___________________________________________________________________________

Other comments

Query #1: Line 53: '530 million infected', it must be many more than this, especially given estimates that 80% of infections are asymptomatic.

Response: Thanks to the reviewer’s observation. As per the suggestion, statistics related to the total number of people infected and diseased with SARS CoV-2 are updated (Lines: 52-54).

“The severity of COVID-19 pandemic is a major concern as it infected 550 million people and caused 6.32 million deaths across the globe [1].” (Lines: 52-53).

___________________________________________________________________________

Query #2: Lines 55-57: Provide a date for which the website was accessed for citing the counts.

Response: As per the suggestion of the reviewer top five countries with the highest numbers of reported cases and deaths are updated in the text and the website from which the data was accessed for citing the count was cited as a new reference [1] (Lines: 54-59).

“The top five countries with the highest numbers of reported cases are the United States of America (8567342082117634), India (4336229443131822), Brazil (3189073330741811), France (2941674028425371), and Germany (2768177525998085); and the top five countries with the highest number of reported deaths are United States of America (1005025993691), Brazil (669530665319), India (524954524323), Russian Federation (380776378168), and Mexico (325487324617) [1] (https://covid19.who.int/).(Lines: 54-59)”.

___________________________________________________________________________

Query #3: Lines 83-84: Ct value and viral load copies/ml are essentially deterministically linked through a conversion formula. I would think the logical point to make here would be citing evidence that Ct value is associated with the severity of the disease.

Response: We thank the reviewer for the suggestion. In our study, RT-PCR for COVID-19 detection is a qualitative type that determines if a sample is positive or negative for SARS-CoV-2 infection. Some studies link the Ct values with the severity by estimating the scores (doi.org/10.1007/s15010-022-01783-1). While other studies have observed No correlation between Ct values and severity of disease or mortality in patients with COVID 19 disease. (doi.org/10.1016/j.ijmmb.2020.10.021) Because of these conflicting reports, in our study, we performed the qualitative RT-PCR indicating a sample was positive or negative. Accordingly, we have cited references in the manuscript. (Line: 79-81).

___________________________________________________________________________

Query #4: Line 88: This sentence suggests vaccine efficacy against INFECTION, but really these vaccines are not that efficacious against infection. They are efficacious against symptomatic disease. That is true for essentially every good vaccine. I would recommend, when referring to vaccine efficacy, always restricting to prevention of symptomatic disease; this was always the primary endpoint of the phase 3 trials. Mary Lou Clements-Mann in a 1998 AIDS Res Hum Retroviruses paper is a classic commentary on how there is often confusion in that people talk about vaccine efficacy against infection, but really almost all vaccine efficacy trials study symptomatic disease endpoints, not infection endpoints.

Response: We thank the reviewer for the correction. Yes, we agree that vaccines protect against the severity of the disease and not against the severity of the infection. However, there are a few studies that show vaccines could be efficacious against infection (10.1016/S1473-3099(21)00690-3). In this study, the authors confirm that COVID-19 vaccination reduces the risk of delta variant infection and also accelerates viral clearance in the context of the delta variant. However, as per the suggestion of the reviewer, we accordingly we have made changes in the manuscript at appropriate places (Lines:81-83).

___________________________________________________________________________

Query #5: Lines 104-105: rephrase to note disease reduction is the benefit, not infection reduction. This issue comes up again in lines 202-203.

Response: Authors would like to thank the reviewer for the comment. As per the suggestion, we have incorporated changes in the manuscript at appropriate places. (Line: 99)

___________________________________________________________________________

Query #6: Line 155: "Patients who were admitted with COVID like syndrome" Why were they excluded? On the surface this sounds like excluding COVID-19 patients.

Response: We thank the reviewer for the comment. Patients who were admitted with COVID-like syndrome such as headache, fever, cough, cold etc., were excluded in our study because they were negative for COVID-19 by RT-PCR (Lines: 141-143).

___________________________________________________________________________

Query #7: Lines 234-236: a typo in stating that 72.7% is 'much higher than 68.9% and 70.2%

Response: We would like to thank the reviewer for the comment. The ‘much higher’ is corrected as ‘slightly higher’ (Lines: 220-221).

__________________________________________________________________________

Query #8: Lines 264-266: 'sequentially higher' is not accurate

Response: Authors would like to thank the reviewer for the comment. The word ‘sequentially higher’ is corrected as ‘substantially higher’ (Line: 244).

___________________________________________________________________________

Reviewer 2 Report

Major comments

 Introduction

-Lines 52-53. They talk about COVID-19 pandemic as a past Public Health concern. I suggest writing as a current issue and show latest figures. Moreover, the citation [2] is not properly used when referring morbility and mortality across the globe since ref. [2] is concerned to Wuhan. A useful citation with update information worldwide (more suitable for the sentence in line 52-53) is:

 ·         Dong E, Du H, Gardner L. An interactive web-based dashboard to track COVID-19 in real time. The Lancet Infectious Diseases. 2020; 20: 533-4.

-Line 58. The website must also be referred in References section, and with a number in brackets in the text. I suggest this:

 ·         WHO. WHO Coronavirus (COVID-19) Dashboard. Available from: https://covid19.who.int/ [date of Access]

-Lines 58-60. Authors must support their epidemiological statement with more than a single study.

-Lines 71-76. The citation [5] is not correct and it not used well anywhere on those lines when referring to the clinical features of SARS-Cov-2 infection. Please use others more suitable. I suggest two:

·         Chen N, Zhou M, Dong X, Qu J, Gong F, Han Y, et al. Epidemiological and clinical characteristics of 99 cases of 2019 novel coronavirus pneumonia in Wuhan, China: a descriptive study. Lancet. 2020; 395: 507-513.

     ·         Hu B, Guo H, Zhou P, Shi ZL. Characteristics of SARS-CoV-2 and COVID-19. Nat Rev Microbiol. 2021; 19: 141-154. doi: 10.1038/s41579-020-00459-7. Erratum in: Nat Rev Microbiol. 2022; 20: 315.

-Line 89. The figure shown (354 vaccines candidates) is according to WHO source as authors stated, but no reference from this institution is included. I suggest this webpage:   COVID-19 vaccine tracker and landscape (who.int)

In addition, the link to https://covid19.trackvaccines.org/ must be included in References section, with the date of Access and the organisation managing such webpage.

-Lines 97-98. Literature referring to COVAXIN vaccine is not correct. Citation [20] is related to Covishield. Instead, they can use the following ones:

·         Sapkal GN, Yadav PD, Ella R, Deshpande GR, Sahay RR, Gupta N, Vadrevu KM, Abraham P, Panda S, Bhargava B. Inactivated COVID-19 vaccine BBV152/COVAXIN effectively neutralizes recently emerged B.1.1.7 variant of SARS-CoV-2. J Travel Med. 2021; 28: taab051.

·         Sharma R, Tiwari S, Dixit A. Covaxin: An overview of its immunogenicity and safety trials in India. Bioinformation. 2021; 17: 840-845.

-Lines 100-102. The use of citation [22] is not correct either. This article refers to an adverse event following COVAXIN vaccination but not to the explaination of its own mechanism of action.

-Line 112. Before “Hence” use a full stop. In addition, rewrite the sentence emphasizing that vaccines were “firstly” or “preferently” administered to that group of patients, because vaccines were actually approved for all the range of ages (except young children); the shortage of vaccines was the reason of prioritizing high-risk population. Furthermore, replace “above” by “older” (also in line 186).

Methods

-My main concern is related to the analyses performed (or rather, not performed). Authors should have done a multivariable exploration with statistical significant explanatory variables.

-The final number of patients included in “Study design” (n=820) shouldn´t be included in this section. It is also mentioned in Results, where it is more suitable.

-The heading “Study sites and sample size” and its content must be included in the first subsection of Methods, “Study design”. The same as “Data collection and case enrolment” and “Ethical consideration”; all the information included is already mentioned in previous headings. Therefore, the authors should merge the subsections into one.

-Line 155. One of the exclusion criteria, COVID like syndrome, is not well explained enough. Is authors referring to COVID like syndrome with a negative test? I guess so, but they should specify it.

-They have to indicate the p-value considered as statistically significant (Statistical Analysis subheading).

Results

-My main concern is regarding a missing comparison among groups (unvaccinated, single-dose vaccinated and fully vaccinated). Just a p-value is shown (line 196) but I suggest the inclusion of a overall table with the values obtained for all the variables in every group, as well as a final column where they show the statistical significance of each comparison.

-Lines 170-173 must be deleted; it is repetitive with the information provided in the previous paragraph.

-Lines 176-177. The final sentence of this paragraph must be deleted; this information is already provided in Methods section. In addition, the version of SPSS provided in Results (version 21.0) differs from that given in Methods (versión 20.0).

-Line 182. The sentence “A total of 820 patients were investigated in this study” has to be deleted.

-Line 182. “520 out of 820” instead of “Out of which 520”. In addition, it is not necessary the mention of males and females (you can infer one of them from the percentage of the other!!); so I suggest citing just one of the figures and the ratio.

-Since most of patients belongs to a certain group of age, we can think about a non-normal distribution, so the authors must use median and interquartile range.

-Figures 2 and 5 are not cited anywhere in the text.

-References [32] and [33] and the subsequent explanation should not be included in this section. The choice of saturation measure must be discussed in the corresponding section, Discussion.

-Lines 197-203. All this paragraph is not a Result but only a discussion. Move it to that section.

-Lines 207-209. These lines include a justification of use of viral markers; please, relocate in Discussion section.

-Line 211. “Higher” instead of “high”. In this paragraph, if authors use mean, they also have to include the standard deviation. Standard deviation is also missed when they show average hospital stay (subheading 3.5, from line 246).

-Lines 220-227; 254-256. All these paragraphs are not results but only discussions. Hence, move it to that section.

-Lines 258-259. These lines include a justification of use of viral markers; please, relocate in Discussion section. 

-Lines 263 and 350. “Higher” instead of “high”. 

-Lines 266-268. These lines include a justification. Move them to Discussion. 

Discussion

-Lines 297-298.They mention the different outcome between the different type of vaccine. This discrimination should be done in this study as well. Is there any difference when patients are fully vaccinated with the same or different type of vaccine? Is there any difference between vaccines? These comparison should have also been performed. 

-Line 337. There is an extra bracket.

Author Response

Dear Dr. Federico,

We sincerely thank the reviewer for their time and for their comments to improve the quality of our article. Please find the point-by-point response to each of the reviewer’s comments below. 

Reviewer-2

Introduction

Query #1: Lines 52-53. They talk about the COVID-19 pandemic as a past Public Health concern. I suggest writing as a current issue and showing latest figures. Moreover, the citation [2] is not properly used when referring morbidity and mortality across the globe since Ref. [2] is concerned to Wuhan. A useful citation with update information worldwide (more suitable for the sentence in line 52-53) is:

Dong E, Du H, Gardner L. An interactive web-based dashboard to track COVID-19 in real-time. The Lancet Infectious Diseases. 2020; 20: 533-4.

Response: We would like to thank the reviewer for the suggestions. As per the suggestions of the reviewer, current statistics related to the total number of people infected and diseased with SARS CoV-2 are updated. Previous Ref. [2] in line 52-53 is replaced with new Ref. [2] (Line 52-53).

“The severity of COVID-19 pandemic is a major concern as it infected 550 million people and caused 6.32 million deaths across the globe [1].”

“1. Organization, W.H. WHO Coronavirus (COVID-19) Dashboard. Available online: https://covid19.who.int/data (accessed on 25 June 2022). (Lines: 613-614)”

___________________________________________________________________________

Query #2: Line 58. The website must also be referred in the References section, and with a number in brackets in the text. I suggest this:

WHO. WHO Coronavirus (COVID-19) Dashboard. Available from: https://covid19.who.int/

[Date of Access]

Response: We thank the reviewer for the suggestion. We have added new Ref. [1] is added to cite the website from which the content has been taken (Line 59).

“1. World Health Organization. WHO Coronavirus (COVID-19) Dashboard. Available online: https://covid19.who.int/data (accessed on 25 June 2022). (Lines: 660-661)”

___________________________________________________________________________

Query #3: Lines 58-60. Authors must support their epidemiological statement with more than a single study.

Response: We thank the reviewer for the suggestion. In this revised version, we have included the supporting evidence to the epidemiological statement with the addition of one more new reference [2] (Lines: 58-59).

“2. Davies, N.G.; Klepac, P.; Liu, Y.; Prem, K.; Jit, M.; group, C.C.-w.; Eggo, R.M. Age-dependent effects in the transmission and control of COVID-19 epidemics. Nat Med 2020, 26, 1205-1211, doi:10.1038/s41591-020-0962-9. (Lines: 662-663)”

___________________________________________________________________________

Query #4: Lines 71-76. The citation [5] is not correct and it is not used well anywhere on those lines when referring to the clinical features of SARS-Cov-2 infection. Please use others more suitable. I suggest two:

Chen N, Zhou M, Dong X, Qu J, Gong F, Han Y, et al. Epidemiological and clinical characteristics of 99 cases of 2019 novel coronavirus pneumonia in Wuhan, China: a descriptive study. Lancet. 2020; 395: 507-513.

Hu B, Guo H, Zhou P, Shi ZL. Characteristics of SARS-CoV-2 and COVID-19. Nat Rev Microbiol. 2021; 19: 141-154. DOI: 10.1038/s41579-020-00459-7. Erratum in: Nat Rev Microbiol. 2022; 20: 315.

Response: We are very thankful for the suggestion. As per the suggestion, we have included two references in this revised version. [3,4] (Lines: 71-73).

“3. Chen, N.; Zhou, M.; Dong, X.; Qu, J.; Gong, F.; Han, Y.; Qiu, Y.; Wang, J.; Liu, Y.; Wei, Y.; et al. Epidemiological and clinical characteristics of 99 cases of 2019 novel coronavirus pneumonia in Wuhan, China: a descriptive study. Lancet 2020, 395, 507-513, doi:10.1016/S0140-6736(20)30211-7.

  1. Hu, B.; Guo, H.; Zhou, P.; Shi, Z.L. Characteristics of SARS-CoV-2 and COVID-19. Nat Rev Microbiol 2021, 19, 141-154, doi:10.1038/s41579-020-00459-7. (Line 664-668)”

___________________________________________________________________________

Query #5: Line 89. The figure shown (354 vaccines candidates) is according to WHO source as authors stated, but no reference from this institution is included. I suggest this webpage: COVID-19 vaccine tracker and landscape (who.int)

In addition, the link to https://covid19.trackvaccines.org/ must be included in the References section, with the date of Access and the organisation managing such webpage.

Response: As suggested by the reviewer, the new Ref. [5] is added to cite the website from which the content has been taken (Line: 91).

  1. World Health Organization. COVID-19 vaccine tracker and landscape. Available online: https://www.who.int/publications/m/item/draft-landscape-of-covid-19-candidate-vaccines (accessed on 26 May 2022). (Line: 669-671)

In addition, one more new reference [6] is also added to cite the website from which the content has been taken (Line: 91).

  1. COVID19 VACCINE TRACKER. COVID19 Vaccine Tracker. Available online: https://covid19.trackvaccines.org/ (accessed on 13 May 2022). (Lines: 625-626).

___________________________________________________________________________

Query #6: Lines 97-98. Literature referring to COVAXIN vaccine is not correct. Citation [20] is related to Covishield. Instead, they can use the following ones:

Sapkal GN, Yadav PD, Ella R, Deshpande GR, Sahay RR, Gupta N, Vadrevu KM, Abraham P, Panda S, Bhargava B. Inactivated COVID-19 vaccine BBV152/COVAXIN effectively neutralises recently emerged B.1.1.7 variant of SARS-CoV-2. J Travel Med. 2021; 28: taab051.\

Sharma R, Tiwari S, Dixit A. Covaxin: An overview of its immunogenicity and safety trials in India. Bio information. 2021; 17: 840-845.

Response: We would like to thank the reviewer for the comment. As per the suggestion of the reviewer, the previous reference [20] is replaced with the new references [7,8] (Line: 93).

  1. Sharma, R.; Tiwari, S.; Dixit, A. Covaxin: An overview of its immunogenicity and safety trials in India. Bio information 2021, 17, 840-845, doi:10.6026/97320630017840.
  2. Sapkal, G.N.; Yadav, P.D.; Ella, R.; Deshpande, G.R.; Sahay, R.R.; Gupta, N.; Vadrevu, K.M.; Abraham, P.; Panda, S.; Bhargava, B. Inactivated COVID-19 vaccine BBV152/COVAXIN effectively neutralises recently emerged B.1.1.7 variant of SARS-CoV-2. J Travel Med 2021, 28, doi:10.1093/jtm/taab051. (Lines: 627-631)”.

___________________________________________________________________________

Query #7: Lines 100-102. The use of citation [22] is not correct either. This article refers to an adverse event following COVAXIN vaccination but not to the explanation of its own mechanism of action.

Response: We thank the reviewer for the correction. The previous reference [22] is replaced with the new references [9] (Line: 93).

“9. INDIAN COUNCIL OF MEDICAL RESEARCH. World Health Organisation approval for COVAXIN - A path breaking moment for India. (Line: 632)”

___________________________________________________________________________

Query #8: Line 112. Before “Hence” use a full stop. In addition, rewrite the sentence emphasising that vaccines were “firstly” or “preferentially” administered to that group of patients, because vaccines were actually approved for all the range of ages (except young children); the shortage of vaccines was the reason of prioritising high-risk population. Furthermore, replace “above” by “older” (also in line 186).

Response: We are very much thankful for the suggestion. and sentence from Line: 112 has been rewritten and modified as per the reviewer suggestion (Lines: 113-117) and also “above” is replaced with “older” (Line:190).

“Mortality due to COVID-19 has been reported to be relatively high in aged (60yrs and above) patients [27], and in particular, those individuals with comorbidities such as diabetes, hypertension, cardiovascular disease, cancer, etc. [28-30]. Therefore, such individuals were categorized as high-risk group and were administered with vaccine on priority. This group was followed by individuals between 45 yrs to 59 yrs age and subsequently for the younger group.  (Lines: 104-109).”

Shortage of vaccines was not the major reason in India at that time. COVISHIELD was made available in sufficient quantities. It is because of susceptibility of people in higher age group and those with comorbidities. Only when vaccination was made open for all adults, then to some extent shortage was felt for short time.

“Majority of the patients (65%) were belonged to 31-60 years of age group with more than 20% of patients in the age group above 60yrs while only <15% patients in the 18-30yrs of age group. (Figure 2).

(Line: 183)”.

___________________________________________________________________________

Methods: 

Query #1: My main concern is related to the analyses performed (or rather, not performed). Authors should have done a multivariable exploration with statistically significant explanatory variables.

Response: We thank the reviewer for suggestion. in our manuscript, we have analysed a total of 820 patients for variables such as oxygen saturation, serum ferritin, clinical features (cough, breathlessness, fever etc.,), duration of hospital stay between unvaccinated, single-dose vaccinated and fully vaccinated patients. Where applicable, the data were analysed using one-way ANOVA with Tukey’s Post Hoc Test. The significant difference was represented by * and *** p-value <0.05 and <0.0001 respectively.

___________________________________________________________________________

Query #2: The final number of patients included in the “Study design” (n=820) shouldn´t be included in this section. It is also mentioned in Results, where it is more suitable.

Response: Authors would like to thank the reviewer for the comment, and as per the reviewer's suggestion number of patients in the study (n=820) has been removed from this section (Methods) and included only in the Results section (Line: 125).

___________________________________________________________________________

Query #3: The heading “Study sites and sample size” and its content must be included in the first subsection of Methods, “Study design”. The same as “Data collection and case enrolment” and “Ethical consideration”; all the information included is already mentioned in previous headings. Therefore, the authors should merge the subsections into one.

Response: We agree with reviewer’s observation. As per the suggestion of the reviewer, we have merged the subsection into one in this revised version of the manuscript (Lines: 116-132).

___________________________________________________________________________

Query #4: Line 155. One of the exclusion criteria, COVID-like syndrome, is not well explained enough. Are authors referring to COVID-like syndrome with a negative test? I guess so, but they should specify it.

Response: We thank the reviewer for the comment. In our study, patients with symptoms like (COVID like) common cold, fever, and cough were excluded based on the negative RT-PCR test report. The same has been mentioned in this revised version of the manuscript (Lines:141-143).

___________________________________________________________________________

Query #5: They have to indicate the p-value considered as statistically significant (Statistical Analysis subheading).

Response: As per the reviewer suggestion p-value is included under the statistical analysis subheading (Line 167).

“A p-value of 0.05 and 0.0001 was considered statistically significant. (Lines: 161-162)”.

___________________________________________________________________________

Results

Query #1: My main concern is regarding a missing comparison among groups (unvaccinated, single-dose vaccinated and fully vaccinated). Just a p-value is shown (line 196), but I suggest the inclusion of an overall table with the values obtained for all the variables in every group, as well as a final column where they show the statistical significance of each comparison.

Response: We express our thanks to the reviewer’s observation. As per the suggestion, we have included an overall table with the values (Mean±SD) obtained for all the variables in every group, along with the significance of each comparison.

Parameter

Unvaccinated Mean±SD (A)

Single Dose Mean±SD (B)

Fully vaccinated Mean±SD (C)

Comparing A with B (p-value)

Comparing A with C (p-value)

Oxygen saturation

89.53%+8.55%

91.94+6.71%

95.31+2.53%

<0.05*

<0.0001***

Serum ferritin (ng/ml)

665.85±557.70

412.23±421.32

282±349.88

<0.0001***

<0.0001***

Query #2: Lines 170-173 must be deleted; it is repetitive with the information provided in the previous paragraph.

Response: As per the reviewer suggestion lines 170-173 have been deleted (Lines: 164-167).

___________________________________________________________________________

Query #3: Lines 176-177. The final sentence of this paragraph must be deleted; this information is already provided in the Methods section. In addition, the version of SPSS provided in Results (version 21.0) differs from that given in Methods (version 20.0).

Response: As per the suggestion, lines 176-177 have been deleted (Lines 175) and version of the SPSS provided in the results is a typo error (typed 21.0 instead 20.0). Since it is already mentioned in the methods (158), we deleted it from the results section (Line: 175).

___________________________________________________________________________

Query #4: Line 182. The sentence “A total of 820 patients were investigated in this study” has to be deleted.

Response: As suggested, the indicated sentence has been deleted in the revised version (Line: 181).

___________________________________________________________________________

Query #5: Line 182. “520 out of 820” instead of “Out of which 520”. In addition, it is not necessary the mention of males and females (you can infer one of them from the percentage of the other!!); so I suggest citing just one of the figures and the ratio.

Response: As per the reviewer’s suggestion, we replaced ‘Out of which 520’ with ‘520 out of 820’ (Line 186) and also, we included only male’s population and male to female sex ratio (Line: 182) in this revised version. 

“The male to female sex ratio was 1.73:1. The mean age of the patients was 50.55+5.35yrs.”

___________________________________________________________________________

Query #6: Since most of patients belongs to a certain group of age, we can think about a non-normal distribution, so the authors must use median and interquartile range.

Response: We thank the reviewer for the suggestion. However, we believe that, Mean values would also represent the same outcome of the study. Consistent with other studies, we preferred to stay with the current statistical analyses. 

___________________________________________________________________________

Query #7: Figures 2 and 5 are not cited anywhere in the text.

Response: We thank the reviewer for the note. In this revised version, figures 2 and 5 are cited in the text (Lines: 185 and 193).

___________________________________________________________________________

Query #8: References [32] and [33] and the subsequent explanation should not be included in this section. The choice of saturation measure must be discussed in the corresponding section, Discussion.

Response: As per the suggestion of the reviewer, references [32] and [33] and the subsequent explanation is moved to the discussion section (Lines: 291-294).

The National Institutes of Health (NIH), USA recommended a target oxygen saturation range of 92-96% for the patients with COVID-19 [32]. Patients with COVID-19 infections usually experience low level of oxygen in their blood [33]. (Lines:291-294).

___________________________________________________________________________

Query #9: Lines 197-203. All this paragraph is not a Result but only a discussion. Move it to that section.

Response: As per the suggestion of the reviewer Lines 197-203 have been moved to the discussion section (Lines: 340-343).

“We observed a slight drop in oxygen saturation between single vs fully vaccinated patients, though one of the very recent studies reported no significant differences in the blood oxygen saturation among vaccinated patients [34]. In another study, Wilder-Smith et al., 2022, reported that vaccination could reduce the risk of even delta variant infection and promote viral clearance [35]”. (Lines: 314-318)”

___________________________________________________________________________

Query #10: Lines 207-209. These lines include a justification for use of viral markers; please, relocate in the Discussion section.

Response: Authors would like to thank the reviewer for the comment Lines 207-209 are moved to the discussion section (Lines: 347-349).

“Recent studies have reported elevated serum ferritin levels in critically ill COVID-19 patients [36]. Ferritin is an iron storage protein involved in regulating immune function and inflammation [37]. (Lines: 323-325)”

___________________________________________________________________________

Query #11: Line 211. “Higher” instead of “high”. In this paragraph, if authors use mean, they also have to include the standard deviation. Standard deviation is also missed when they show average hospital stay (subheading 3.5, from line: 246).

Response: As per the reviewer's suggestions in Line 204 word “high” is replaced with “higher”, and the standard deviation is included in the same paragraph and under the subheading 3.5 (Lines: 233-240).

___________________________________________________________________________

Query #12: Lines 220-227; 254-256. All these paragraphs are not results but only discussions. Hence, move it to that section.

Response: As per the suggestions of the reviewer, lines 220-227 and 254-256 are moved to the discussion section (Lines: 365-372) and (Lines: 396-398) respectively.

“A study reported that the fatal outcomes of COVID-19 are in part due to unusually high cytokines, collectively known as “cytokine storm”. The severity of COVID-19 not only depends on the cytokine storm but also on the ferritin levels [39]. One of the studies reported that patients with very severe COVID-19 exhibited higher serum ferritin levels than in severe COVID-19 individuals [38]. Gomez-Pastora et al., (2020) reported that ferritin concentration above approx. 400ng/ml is a risk factor for progression to severe disease [40]. (Lines: 341-347)”

Similar studies conducted elsewhere have shown the efficacy of vaccines in reducing the severity of COVID-19 and the length of the hospital stay [41] [74] [75] [76] [77]. (Lines; 356-358).

___________________________________________________________________________

Query #13: Lines 258-259. These lines include a justification of the use of viral markers; please, relocate to the Discussion section. 

Response: Authors would like to thank the reviewer for the comment, and as per the suggestions of the reviewer, Lines 258-259 are removed from the results section and moved to the discussion section.

“Diabetes mellitus, hypertension, ischemic heart diseases and chronic kidney diseases were the most common comorbidities reported by COVID-19 patients [42]. (Lines: 362-363)”

___________________________________________________________________________

Query #14: Lines 263 and 350. “Higher” instead of “high”. 

Response: Authors would like to thank the reviewer for the comment, and as per the suggestions of the reviewer in Line 263 and 350 words, “high” is replaced with “higher” (Line 245 and 329).

___________________________________________________________________________

Query #15: Lines 266-268. These lines include a justification. Move them to Discussion.

Response: Authors would like to thank the reviewer for the comment, and Lines: 266-268 have been deleted from the results section (Line: 248).

___________________________________________________________________________

Discussion

Query #1: Lines 297-298.They mention the different outcome between the different types of vaccine. This discrimination should be done in this study as well. Is there any difference when patients are fully vaccinated with the same or different type of vaccine? Is there any difference between vaccines? This comparison should have also been performed.

Response: We sincerely thank the reviewer for the suggestion. We wish to bring to the kind notice that, during the study period, the widely available and accepted vaccine in India was COVISHIELD. All of the patients vaccinated in our study were vaccinated with COVISHIELD. Therefore, we could not find inter-vaccination differences among the patients included in the study. The same has been mentioned at the appropriate place in this revised version of the manuscript.

___________________________________________________________________________

Query #2: Line 337. There is an extra bracket.

Response: Authors would like to thank the reviewer for the comment, and as per the suggestion of the reviewer extra bracket has been removed (Line: 323).

___________________________________________________________________________

Round 2

Reviewer 1 Report

Generally the revision was responsive to statistical comments.

The one exception was Query #2, on implementing adjustment for confounders in analyses for main results that are reported.  It appeared that the revision used unadjusted analyses for main results reporting?  If that is so, would recommend stating that more directly, and in the Discussion noting that a limitation is that confounders were not adjusted for.

It would be better to do the analyses with adjustment for confounders and report those results.

Author Response

We sincerely thank the reviewer for the valuable comment and the recommendation. At this point, as per the suggestion, we have included a statement in the conclusion section.  

"Therefore, the current study requires that outcomes among unvaccinated, single-dose and fully vaccinated patients be adjusted for the above potential confounders."  (Lines: 383-385).

The changes made in the manuscript are highlighted in blue colour.

Thank You,

Best Regards,

Ravindra P V

Reviewer 2 Report

I would like to thank authors for considering most of my concerns. Anyway, I have two left that need to be clarified yet: I cannot find the answer to Query 5, and other major question arosen is if authors have correctly included the references (i.e, after answering query 8, cites 32 and 33 are between cites 53 and 54).

Author Response

We sincerely thank the reviewer for the suggestion and correction.

In this revised version we have corrected the reference (Lines: 86-87), and also rearranged all the references in the order.

The changes made to the manuscript have been highlighted in blue colour. 

Thank You,

Best Regards,

Ravindra